# Growing Visual Generative Capacity for Pre-Trained MLLMs

## Abstract

Multimodal large language models (MLLMs) extend the success of language models to visual understanding, and recent efforts have sought to build *unified MLLMs* that support both understanding and generation. However, constructing such models remains challenging: hybrid approaches combine continuous embeddings with diffusion or flow-based objectives, producing high-quality images but breaking the autoregressive paradigm, while pure autoregressive approaches unify text and image prediction over discrete visual tokens but often face trade-offs between semantic alignment and pixel-level fidelity. In this work, we present **Bridge**, a pure autoregressive unified MLLM that augments pre-trained visual understanding models with generative ability through a Mixture-of-Transformers architecture, enabling both image understanding and generation within a single next-token prediction framework. To further improve visual generation fidelity, we propose a semantic-to-pixel discrete representation that integrates compact semantic tokens with fine-grained pixel tokens, achieving strong language alignment and precise description of visual details with only a 7.9% increase in sequence length. Extensive experiments across diverse multimodal benchmarks demonstrate that Bridge achieves competitive or superior results in both understanding and generation benchmarks, while requiring less training data and reduced training time compared to prior unified MLLMs.

## 1 Introduction

Inspired by the success of large language models (LLMs)(Bai et al., 2023a; Brown, 2020; Radford et al., 2019; Google et al., 2023; Touvron et al., 2023a;b), multimodal large language models (MLLMs)(Liu et al., 2023a; 2024c) have been developed to jointly process and understand visual and textual information within a next-token prediction framework. Leveraging pre-trained visual encoders such as CLIP (Radford et al., 2021) or SigLIP (Zhai et al., 2023; Tschannen et al., 2025), existing approaches typically project the extracted visual features into the latent space of the LLM and fine-tune the model to interpret visual signals through these embeddings. Despite their success across a wide range of visual understanding applications, these kinds of MLLMs are not truly "multimodal", as they can only **understand** but **not generate** visual signals. This limitation not only prevents them from generating images as desired by users, but also hinders their ability to reason through the visual modality (Chern et al., 2025) when tackling more complex multimodal tasks.

To address these limitations, unified MLLMs have been designed to jointly learn visual understanding and generation within a single system. Broadly, existing unified MLLMs fall into two categories. *Hybrid MLLMs* (Deng et al., 2025; Zhou et al., 2024; Xie et al., 2025b) represent visual tokens with continuous embeddings and incorporate generative objectives such as diffusion or flow matching, generating images through an iterative refinement process. In contrast, *pure autoregressive MLLMs* (Han et al., 2025; Wu et al., 2024b; Wang et al., 2024b; Ma et al., 2025a; Chen et al., 2025c) rely on discrete visual tokens encoded by vector-quantized (VQ) tokenizers (Van Den Oord et al., 2017; Esser et al., 2021) and are trained solely with negative log-likelihood loss, thereby following a clean next-token prediction paradigm for both language and vision generation.

Both understanding MLLMs and unified MLLMs are typically initialized from pure language LLMs in order to inherit strong linguistic capabilities. However, while unified MLLMs is functionally a superset of understanding MLLMs, most existing unified MLLMs do not inherit the visual understand-

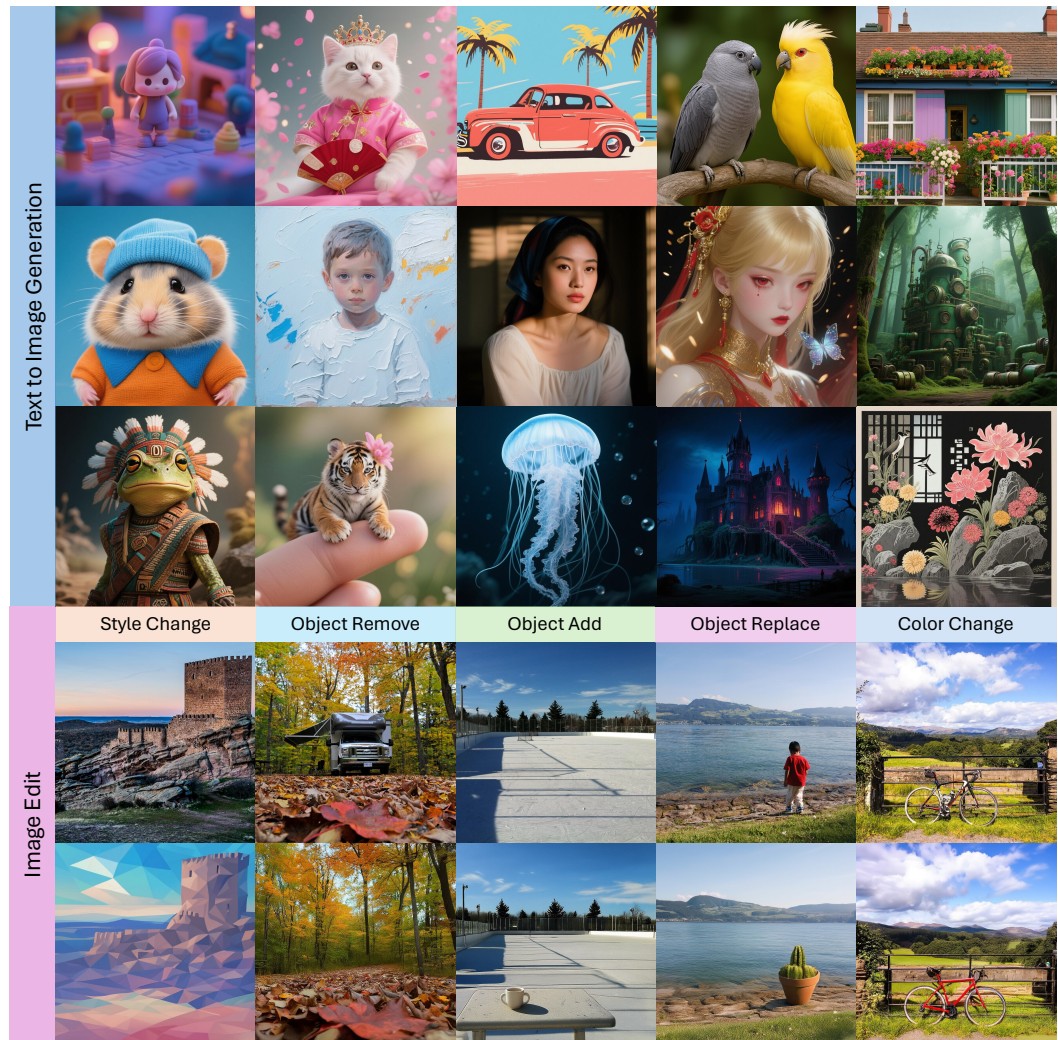

Figure 1: **Qualitative Results** on text-to-image generation and image editing tasks.

ing ability already established in understanding MLLMs. This gap can be attributed to differences in architectural choices and training objectives. For instance, unified MLLMs sometimes discard the pre-trained visual encoders (Han et al., 2025; Wu et al., 2024b) used in understanding-oriented MLLMs, or employ non-autoregressive generative objectives (Deng et al., 2025; Zhou et al., 2024; Chen et al., 2025a) that deviate substantially from the training paradigm of understanding-oriented models.

In this paper, we propose **B**i-modality **R**outing v**i**a **D**ual-branch **G**enerative **E**xperts (**Bridge**), a pure autoregressive unified MLLM that performs both image understanding and generation through the same next-token prediction framework. Unlike prior unified MLLMs, Bridge is directly built upon existing understanding-oriented MLLMs, thereby inheriting their strong visual comprehension capabilities. Crucially, by adopting a dual-branch modeling design, our approach preserves the inherited model's visual understanding ability while enabling generative capacity, ensuring that comprehension is not compromised. This design reduces dependence on large-scale, high-quality visual understanding datasets, while also decreasing training time and enhancing overall training efficiency.

To further enhance visual generation quality, we propose a *semantic-to-pixel discrete representation* for images. This representation integrates two complementary levels of tokenization: compact, high-level semantic tokens that capture global structure and meaning, and fine-grained pixel tokens that

preserve detailed visual information such as textures and edges. By combining semantic abstraction with pixel-level precision, our discrete visual representation achieves both strong alignment with language modeling and accurate reconstruction of visual details. Remarkably, this richer representation requires only 7.9% increase in per-image token length compared to only using pixel tokens, yet it delivers significantly improved generation fidelity across diverse tasks.

To summarize, our key contributions are as follows:

- We propose **Bridge**, a pure autoregressive unified MLLM that expands pre-trained MLLMs with visual generative capacity while strictly preserving their inherited visual understanding ability. This design enables efficient unification of multimodal understanding and generation under a single next-token prediction framework.

- We introduce a *semantic-to-pixel discrete visual representation*, which combines compact high-level semantic tokens with fine-grained pixel tokens. This dual-level representation provides strong semantic alignment with language models while preserving detailed visual fidelity, significantly boosting generation quality with minimal increase in sequence length.

- We conduct extensive experiments across multimodal benchmarks, showing that our method achieves competitive or superior performance in both understanding and generation tasks, while requiring less training data, shorter training time, and overall improved efficiency compared to prior unified MLLMs.

## 2 RELATED WORK

**Unified Multimodal Large Language Models.** Recent advances in multimodal large language models (MLLMs) aim to move beyond visual understanding to also support visual generation within a unified framework. Existing approaches can be broadly divided into two families. The first employs continuous visual embeddings combined with diffusion or flow-based objectives for image generation, as in Transfusion (Zhou et al., 2024), BAGEL (Deng et al., 2025), Show-o2 (Xie et al., 2025b), JanusFlow (Ma et al., 2025c), and ILLUME (Wang et al., 2024a). While capable of producing high-quality synthesis, these methods break the clean autoregressive paradigm, thereby complicating training and integration. The second line of work adopts discrete vision tokens produced by VQ-based tokenizers (Van Den Oord et al., 2017; Esser et al., 2021), enabling unified next-token prediction across both text and image tokens. Within this family, models such as Emu3 (Wang et al., 2024b), Chameleon (Team, 2024), and Janus (Wu et al., 2025c; Chen et al., 2025c) achieve more elegant unification, but their reliance on pixel-level VQ tokenizers (Esser et al., 2021; Sun et al., 2024) often limits semantic alignment with language models and weakens visual comprehension (Han et al., 2025). Other variants, such as Tar (Han et al., 2025), UniTok (Ma et al., 2025a), and VILA-U (Wu et al., 2024b), attempt to bridge this gap through semantic quantization, but still suffer from reduced pixel-level fidelity. A few recent works (Lin et al., 2025; Pan et al., 2025; Wu et al., 2025d) attempt to extend pre-trained MLLMs with generative capability, yet they heavily depend on external diffusion or flow-based decoders, and thus cannot be considered truly unified MLLMs. In contrast, Bridge relies solely on next-token prediction for image generation, without the need for any external generative models.

## 3 METHOD

In this section, we present the design of our pure autoregressive unified MLLM, **Bridge**. We begin with the architecture of Bridge in Section 3.1, describing how it is built on top of a pre-trained understanding MLLM. In Section 3.2, we introduce our semantic-to-pixel discrete visual representation, which enhances visual generation quality significantly. We then detail the data usage in Section 3.3, and finally outline the full training procedure in Section 3.4.

### 3.1 ARCHITECTURE

Bridge is built upon a pre-trained decoder-only MLLM. We adopt InternVL3 (Zhu et al., 2025) with Qwen2.5 (Qwen et al., 2025) model architecture as the backbone, selected for its competitive performance in both language modeling and visual understanding. To preserve these capabilities,

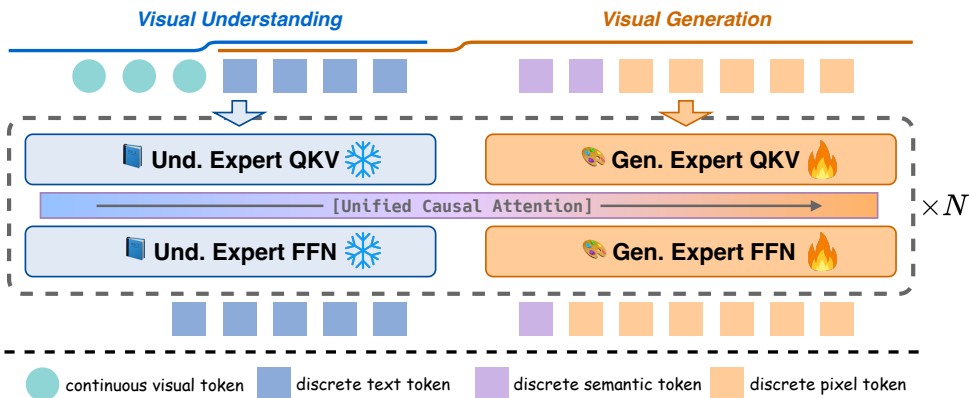

Figure 2: **Method overview.** Bridge adopts a Mixture-of-Transformers (MoT) architecture with two experts: a frozen understanding (Und.) expert for text and visual understanding tokens, and a newly trained generation (Gen.) expert for visual generation tokens. Both experts share unified causal attention across all tokens. Visual generation representation are constructed by concatenating short semantic token sequences with longer pixel token sequences, which are modeled jointly with text tokens under a unified next-token prediction objective. Semantic tokens serve as a bridge between text and pixel modalities, substantially improving visual generation quality.

we keep a complete copy of the InternVL3 model, including its continuous vision encoder, frozen within Bridge. This ensures that the strong language modeling and visual understanding ability of the underlying MLLM remains intact.

To endow the model with visual generation capacity, we introduce additional trainable modules. Unlike prior works such as MetaQueries (Pan et al., 2025) or Qwen-Image (Wu et al., 2025a), which treat the MLLM as a frozen text encoder and rely on external diffusion transformers (Peebles & Xie, 2023; Esser et al., 2024) for image synthesis, our goal is to construct a unified MLLM capable of both understanding and generating multimodal information entirely within the next-token prediction paradigm.

To achieve this, we leverage a Mixture-of-Transformers (MoT) design (Liang et al., 2024; Shi et al., 2024; Deng et al., 2025), as illustrated in Figure 2. Specifically, we copy the LLM backbone to create a generation expert and combine it with the frozen understanding expert. In contrast to Mixture-of-Experts (MoE) architectures, where only the feed-forward networks differ across experts, both the understanding and generation experts in Bridge are implemented as complete transformer blocks with separate parameter sets. The two experts interact at every attention layer, where their tokens are concatenated and processed jointly using standard causal masking, enabling seamless integration of understanding and generation within a unified autoregressive framework.

Hard routing is employed to dispatch tokens between the two experts. Text tokens are passed directly into the understanding expert. Images for understanding tasks are first encoded by the continuous vision encoder inherited from the backbone MLLM and then fed into the understanding expert. In contrast, image generation tokens are routed to the generation expert. During training, ground-truth images are converted into discrete visual tokens and provided to the generation expert for loss computation. At inference, the generation expert autoregressively predicts each discrete visual token, which is then fed back into the same expert to generate the next token.

### 3.2 SEMANTIC-TO-PIXEL DISCRETE VISUAL REPRESENTATION

A central challenge in pure autoregressive unified MLLMs lies in designing effective visual representations for generation. Most existing approaches (Wang et al., 2024b; Team, 2024; Wu et al., 2025c; Chen et al., 2025c) rely on pixel-level tokens produced by VQGANs (Esser et al., 2021). While these tokens capture fine-grained details, their heavy focus on low-level reconstruction makes them poorly aligned with language tokens, often leading to suboptimal image generation quality. In contrast, semantic-level tokens (Han et al., 2025; Wu et al., 2024b), which are encoded by text-

aligned visual encoders, exhibit strong alignment with language but lack sufficient precision for representing detailed visual content.

To combine the strengths of both representations, Bridge adopts a *semantic-to-pixel discrete visual representation*. Each image is represented as a sequence that begins with semantic tokens and is followed by pixel tokens:

$$\texttt{<BOI>} \texttt{<SEM}_0\texttt{>} \texttt{<SEM}_1\texttt{>} \dots \texttt{<PIX}_0\texttt{>} \texttt{<PIX}_1\texttt{>} \dots \texttt{<EOI>},$$

where $\texttt{<BOI>}$ and $\texttt{<EOI>}$ denote special tokens marking the beginning and end of an image, $\texttt{<SEM}_i\texttt{>}$ represents the $i$-th semantic token, and $\texttt{<PIX}_i\texttt{>}$ represents the $i$-th pixel token.

The semantic tokens, placed at the front of the sequence, provide high-level structure and holistic information about the image. Their strong alignment with language tokens makes them easier to generate and helps bridge the modality gap. The subsequent pixel tokens supply fine-grained details necessary for accurate reconstruction. Although pixel tokens alone remain difficult to be modeled, the presence of preceding semantic tokens reduces this difficulty, aligning the entire sequence more effectively with text and thereby improving generation quality. This design is conceptually similar to the Chain-of-Thought (CoT) mechanism (Wei et al., 2022) in LLMs, where intermediate reasoning steps improve the final result.

Concretely, we employ TA-Tok (Han et al., 2025) as the semantic encoder and LlamaGen-VQGAN (Sun et al., 2024) as the pixel encoder. Instead of using TA-Tok's de-tokenizer, we rely solely on the LlamaGen-VQGAN de-tokenizer to reconstruct images from the generated pixel tokens. Importantly, we find that a short, coarse semantic sequence is sufficient to capture holistic image semantics and connect text and pixel tokens. Thus, we use $3\times$ spatially downsampled semantic tokens from TA-Tok, resulting in 81 semantic tokens. When combined with 1024 pixel tokens for representing $512 \times 512$ images, this increases the total token length by less than 10%, while yielding significantly improved visual generation fidelity, as demonstrated in Section 4.2.

### 3.3 DATA CURATION

Our training data consists of image-to-text, text-to-image and interleaved multimodal datasets. Among them, our main goal is to collect and filter large-scale high-quality data for visual generation tasks. **(1) Filtering and recaptioning.** We first apply the aesthetic score and resolution filtering to the webdataset LAION-5B (Schuhmann et al., 2022), where images with an aesthetic score >5 and resolution >512px will be preserved. Then we leverage open-sourced VLMs (Bai et al., 2025; Zhu et al., 2025) to generate more accurate captions for these images. To enhance the model's generation capability on text rendering and human face, we also apply text and human face detection pipeline to curate a subset. **(2) Synthetic Images.** To further improve the model's generation quality, we collect open-sourced datasets generated by GPT-4o-Image (Chen et al., 2025a;b; Ye et al., 2025a; Wang et al., 2025b) and FLUX (Han et al., 2025). Besides, we also leverage Seedream 3.0 (Gao et al., 2025) to curate million scale text-to-image datasets using prompts from other datasets (Deng et al., 2009; DrawThingsAI, 2024) and users (Sun et al., 2023; Egan et al., 2024). Due to space limit, we put the dataset details in the Appendix Section C.

### 3.4 TRAINING PROCEDURE

Since Bridge represents images entirely with discrete tokens, we train it using a unified negative log-likelihood loss across all modalities. This formulation allows text and image tokens to be modeled consistently under the same autoregressive objective. To fully exploit diverse multimodal data, we adopt a multi-stage training procedure inspired by prior LLMs and MLLMs. Below, we describe the details of each stage.

**Stage 1: Unified Multimodal Pre-training.** The objective of this stage is to establish robust multimodal generation capability. We pre-train Bridge on large-scale heterogeneous datasets that include text-to-image pairs (Schuhmann et al., 2022), interleaved multimodal documents (Li et al., 2024c; Qu et al., 2025a), and a small proportion of image-to-text data (Gu et al., 2024; Yu et al., 2024c). During this stage, all input images, including those from image-to-text samples, are routed through the newly introduced generation encoder and processed by the generation expert. In the latter case, textual outputs are still produced by the frozen understanding expert but are conditioned on latent

Table 1: **Quantitative Results on Visual Understanding Benchmarks.** Models without a specified base (M)LLM are trained from scratch.

| Model | Base (M)LLM | POPE↑ | MME-P↑ | MME-C↑ | MMB↑ | SEED↑ | MMMU↑ |
|---|---|---|---|---|---|---|---|
| *Understanding Only Models* | | | | | | | |
| LLaVA-v1.5 | Vicuna-7B | 85.9 | 1511 | - | 64.3 | 58.6 | 35.4 |
| Qwen-VL | Qwen-7B | - | 1488 | - | 60.6 | 58.2 | - |
| LLaVA-NeXT | Vicuna-7B | 86.5 | 1519 | - | 67.4 | 64.7 | 35.1 |
| DeepSeek-VL | DeepSeek-7B | 88.1 | - | - | 73.2 | 70.4 | 36.6 |
| LLaVA-OV | Qwen2-7B | 87.2 | 1580 | 418 | 80.8 | 75.4 | 48.8 |
| *Unified Models* | | | | | | | |
| ILLUME | Vicuna-7B | **88.5** | 1445 | - | 65.1 | 72.9 | 38.2 |
| Chameleon | - | - | - | - | - | - | 22.4 |
| LWM | LLaMA2-7B | 75.2 | - | - | - | - | - |
| Emu3 | - | 85.2 | - | - | 58.5 | 68.2 | 31.6 |
| Liquid | GEMMA-7B | 81.1 | 1119 | - | - | - | - |
| UniTok | LLaMA2-7B | 83.2 | 1448 | - | - | - | - |
| VILA-U | LLaMA2-7B | 85.8 | 1402 | - | - | 59.0 | - |
| Janus-Pro | DeepSeek-7B | 87.4 | 1567 | 260 | 79.2 | 72.1 | 41.0 |
| TokenFlow-XL | Qwen-2.5-14B | 87.8 | 1551 | 371 | 76.8 | 72.6 | 43.2 |
| MetaMorph | LLaMA-3.1-8B | - | - | - | 75.2 | 71.8 | 41.8 |
| Tar | Qwen2.5-7B | 87.8 | 1571 | 355 | 74.4 | 73.0 | 39.0 |
| Show-o2 | Qwen2.5-7B | - | 1621 | - | 79.3 | 69.8 | 48.9 |
| BAGEL | Qwen2.5-7B | - | 1687 | - | **85.0** | - | 55.3 |
| LMFusion | LLaVA-NeXT-8B | - | 1604 | - | 72.1 | 72.5 | 41.7 |
| MetaQuery-XL | LLaVA-NeXT-8B | - | 1685 | - | 83.5 | 76.9 | **58.6** |
| UniWorld-V1 | Qwen2.5-VL-7B | - | - | - | 83.5 | - | **58.6** |
| BLIP3-o | Qwen2.5VL-7B | - | 1683 | 647 | 83.5 | **77.5** | 50.6 |
| **Bridge (Ours)** | InternVL3-8B | 88.4 | **1730** | **677** | 84.4 | 77.4 | 57.4 |

representations provided by the generation expert. This setup encourages the generation expert to learn rich and discriminative visual features, thereby improving downstream tasks that require image-conditioned generation, such as image editing and inpainting. The full pre-training stage uses 410M text-to-image pairs for visual generation, 57M image-to-text pairs for visual understanding, and 29M interleaved multimodal samples.

**Stage 2: Continued Pre-training.** In this stage, we focus on further enhancing the visual generation ability of Bridge by training on high-quality text-to-image datasets (Han et al., 2025; Schuhmann et al., 2022). Compared to the large but diverse corpus used in Stage 1, this data has higher aesthetic quality and includes more challenging cases, such as images with OCR contents or human faces. This stage exposes the model to approximately 60M multimodal examples (partially filtered from the pre-training corpus), enabling it to refine generation quality while improving robustness on visually complex and semantically demanding scenarios.

**Stage 3: Supervised Fine-tuning.** The supervised fine-tuning (SFT) stage uses the smallest amount of data, but of the highest quality. We leverage carefully curated text-to-image datasets (Deng et al., 2009; Sun et al., 2023; Egan et al., 2024; DrawThingsAI, 2024; Chen et al., 2025a;b; Ye et al., 2025a) and instructional image editing data (Wang et al., 2025b; Wu et al., 2025d; Wei et al., 2024; Yu et al., 2024a) to align the model's outputs with human preferences. In total, this stage involves approximately 28M training samples, providing precise guidance for controllable and instruction-following generation.

Additional training details and hyper-parameters can be found in Appendix Section D.

## 4 EXPERIMENTS

In all experiments, we use InternVL3-8B (Zhu et al., 2025) as the base MLLM. Our semantic-to-pixel discrete representation is built upon two components: the visual encoder of TA-Tok (Han et al., 2025) for semantic tokens and LlamaGen-VQGAN (Sun et al., 2024) for pixel tokens. The TA-Tok encoder takes images of size $384 \times 384$ as input, encodes and pools the features, and quantizes them

Table 2: **Quantitative Results on Text-to-image Generation Benchmarks.** † refers to the methods using prompt augmentation, *e.g.*, LLM rewriter or self-CoT (Wei et al., 2022).

| Method | DPG Bench | | | GenEval | | | WISE | | |
|---|---|---|---|---|---|---|---|---|---|
| | Entity | Relation | Overall↑ | Two Obj. | Color Attr. | Overall↑ | Time | Space | Overall↑ |
| *Generation Only Model* | | | | | | | | | |
| SDXL | 82.43 | 86.76 | 74.65 | 0.74 | 0.23 | 0.55 | 0.48 | 0.47 | 0.43 |
| Playground v2.5 | 82.59 | 84.08 | 75.47 | - | - | - | 0.58 | 0.55 | 0.49 |
| Hunyuan DiT | 80.59 | 74.36 | 78.87 | - | - | - | - | - | - |
| DALLE3 | 89.61 | 90.58 | 83.50 | 0.87 | 0.45 | 0.67 | - | - | - |
| SD3-Medium | 91.01 | 80.70 | 84.08 | 0.94 | 0.60 | 0.74 | 0.44 | 0.48 | 0.42 |
| SANA-1.5 | - | - | 84.70 | 0.93 | 0.65 | 0.81 | - | - | - |
| NextStep-1 | - | - | 85.28 | - | - | 0.63 / 0.73† | 0.54 | 0.61 | 0.54 / **0.79**† |
| *Unified Model* | | | | | | | | | |
| Chameleon | - | - | - | - | - | 0.39 | - | - | - |
| LWM | - | - | - | 0.41 | 0.15 | 0.47 | | | |
| Emu3 | 86.68 | 90.22 | 80.60 | 0.71 | 0.21 | 0.54 / 0.66† | 0.45 | 0.48 | 0.39 |
| SEED-X-13B | - | - | - | 0.58 | 0.14 | 0.49 | - | - | - |
| Transfusion | - | - | - | - | - | 0.63 | - | - | - |
| ILLUME | - | - | - | 0.86 | 0.28 | 0.61 | - | - | - |
| Janus-Pro-7B | 88.90 | 89.32 | 84.19 | 0.89 | 0.66 | 0.80 | 0.37 | 0.49 | 0.35 |
| Tar-7B | 88.62 | 93.98 | 84.19 | 0.92 | 0.65 | 0.84 | - | - | - |
| Show-o2-7B | 91.78 | 91.81 | **86.14** | 0.87 | 0.62 | 0.76 | - | - | - |
| MetaQuery-XL | - | - | 82.05 | - | - | 0.80† | 0.55 | 0.62 | 0.55 |
| BAGEL | - | - | - | 0.94 | 0.63 | 0.82 / **0.88**† | 0.55 | 0.68 | 0.52 / 0.70† |
| UniWorld-V1 | - | - | - | 0.93 | 0.70 | 0.80 | 0.55 | 0.73 | 0.55 |
| BLIP3-o-8B | - | - | 81.60 | - | - | 0.84 | - | - | 0.62 |
| **Bridge (Ours)** | 90.10 | 92.27 | 85.51 | 0.93 | 0.66 | 0.74 / 0.82† | 0.56 | 0.65 | 0.53 / 0.69† |

Table 3: **Comparison results on ImgEdit.** Bridge achieves the best overall performance.

| Model | Add | Adjust | Extract | Replace | Remove | Background | Style | Hybrid | Action | Overall↑ |
|---|---|---|---|---|---|---|---|---|---|---|
| Instruct-P2P | 2.45 | 1.83 | 1.44 | 2.01 | 1.50 | 1.44 | 3.55 | 1.20 | 1.46 | 1.88 |
| AnyEdit | 3.18 | 2.95 | 1.88 | 2.47 | 2.23 | 2.24 | 2.85 | 1.56 | 2.65 | 2.45 |
| UltraEdit | 3.44 | 2.81 | 2.13 | 2.96 | 1.45 | 2.83 | 3.76 | 1.91 | 2.98 | 2.70 |
| Step1X-Edit | 3.88 | 3.14 | 1.76 | 3.40 | 2.41 | 3.16 | 4.63 | 2.64 | 2.52 | 3.06 |
| BAGEL | 3.56 | 3.31 | 1.70 | 3.30 | 2.62 | 3.24 | 4.49 | 2.38 | 4.17 | 3.20 |
| UniWorld-V1 | 3.82 | 3.64 | 2.27 | 3.47 | 3.24 | 2.99 | 4.21 | 2.96 | 2.74 | 3.26 |
| **Bridge (Ours)** | 3.49 | 2.64 | 2.93 | 3.45 | 3.48 | 3.45 | 4.14 | 3.09 | 3.85 | **3.39** |

into 81 discrete tokens selected from a codebook of size 65,536. The LlamaGen-VQGAN encoder uses a downsampling ratio of 16, producing $32 \times 32 = 1024$ pixel tokens from $512 \times 512$ images, drawn from a codebook of size 16,384. Concatenating the 81 semantic tokens with the 1024 pixel tokens yields a total of 1105 tokens per image. During training, we consistently resize and center-crop images to $512 \times 512$ before feeding them into the model. Besides, to further enhance the visual quality, we also develop an *optional* upscale module using Lumina-Accessory (Team, 2025), raising the output resoluiton to 1024px.

## 4.1 MAIN RESULTS

**Visual Understanding.** We evaluate Bridge on a suite of visual understanding benchmarks, including POPE (Li et al., 2023b), MME (Fu et al., 2024), MMBench (Liu et al., 2023b), Seed-Bench (Li et al., 2023a), and MMMU (Yue et al., 2024). As reported in Table 1, thanks to the strong visual understanding ability inherited from the base MLLM, Bridge achieves state-of-the-art or near state-of-the-art results across these benchmarks. Notably, even on datasets where Bridge does not rank first, its performance remains very close to the best reported results, demonstrating the robustness of its inherited understanding capacity.

**Text-to-image Generation.** We evaluate visual generation performance on three commonly used text-to-image benchmarks, including GenEval (Ghosh et al., 2023), DPG Benchmark (Hu et al., 2024), and WISE (Niu et al., 2025). Results are reported in Table 2. Our Bridge achieves consistently strong results across all three benchmarks. On DPG Bench, it obtains an overall score of 85.51, surpassing most prior unified model and generation-only methods. On GenEval, Bridge achieves

Table 4: **Visual Representation Comparison.** Sem: Semantic token. Pix: Pixel token.

| #Sem. | #Pix. | GenEval | DPG | ImgEdit |
|---|---|---|---|---|
| 0 | 1024 | 0.48 | 71.7 | 2.93 |
| 729 | 0 | 0.57 | 73.1 | 3.16 |
| 729 | 1024 | 0.60 | 73.4 | **3.35** |
| 81 | 0 | 0.56 | 73.7 | 2.82 |
| 81 | 1024 | **0.61** | **75.8** | 3.33 |

Table 5: **MLLM Architecture Comparison.** Und. denotes the harmonic mean of understanding benchmarks including MME, MMBench, SEED Bench, POPE, and MMMU.

| Arch. | Und. | GenEval | DPG |
|---|---|---|---|
| Dense | 90.1 | 0.53 | **77.2** |
| MoT | **108.0** | **0.63** | 76.3 |

Table 6: **Token Routing Comparison.** Visual understanding and generation performance under different token routing schemes. In the Token Routing columns, **Und.** refers to the understanding expert and **Gen.** refers to the generation expert.

| Token Routing | | Und. Benchmarks | | | | | | Gen. Benchmarks | |
|---|---|---|---|---|---|---|---|---|---|
| Und. Img | Text | MMBench | MME-P | MME-C | SEED | POPE | MMMU | GenEval | DPG |
| Und. | Und. | **84.3** | **1730** | **677** | **77.4** | 88.4 | **57.4** | **0.63** | 76.3 |
| Gen. | Und. | 80.9 | 1621 | 541 | 71.0 | **88.6** | 52.6 | 0.60 | 75.6 |
| Und. | Gen. | 79.0 | 1479 | 564 | 75.2 | 85.3 | 46.0 | 0.59 | 77.1 |
| Gen. | Gen. | 74.5 | 1228 | 475 | 71.4 | 80.2 | 42.1 | 0.53 | **77.2** |

0.82 overall score, which is competitive with recent state-of-the-art unified models like BAGEL. On WISE, our model delivers a solid 0.69, reaching the best reported numbers (0.70) of unified models. These results demonstrate that Bridge achieves highly competitive performance across diverse evaluation settings.

**Instructional Image Editing.** We leverage the ImgEdit benchmark (Ye et al., 2025b) to systematically evaluate the instructional image editing performance of Bridge. Comparisons are made against a series of specialist editing models (Brooks et al., 2023; Yu et al., 2024a; Zhao et al., 2024; Liu et al., 2025) as well as unified MLLMs (Deng et al., 2025; Lin et al., 2025). Table 3 summarizes the results. As shown, Bridge not only surpasses the leading editing specialist model Step1X-Edit (Liu et al., 2025), but also outperforms recent unified MLLMs such as BAGEL (Deng et al., 2025) and UniWorld-V1 (Lin et al., 2025). In particular, Bridge achieves significant gains on the *Extract*, *Remove*, *Background*, and *Hybrid* categories, underscoring its ability to follow fine-grained instructions while preserving global consistency.

**Visualization.** As shown in Figure 1, our model can generate a wide range of images including cartoon-style scenes, fantasy illustrations, photorealistic portraits, stylized characters and animals *etc.* In addition, it also supports image editing tasks such as style transfer, object remove or addition, and replacement. For more visualization, please refer to our Appendix Section F.

## 4.2 ABLATION STUDIES

In this section, we present ablation studies to examine several key design choices of our method. Unless otherwise noted, all experiments use 25M high-quality text-to-image samples to establish the visual generation capability of the model. For ablations that also require training on visual understanding and language modeling, we additionally incorporate image understanding and text-only data, maintaining a sampling ratio of $6 : 2 : 1$ across text-to-image, image understanding, and text data.

**Effectiveness and Efficiency of Semantic-to-pixel Visual Representation.** In Bridge, visual representations are constructed by sequentially combining semantic tokens with pixel tokens. To assess the effectiveness of this design, we compare several alternatives, including using only pixel tokens, only semantic tokens, or varying semantic sequence lengths. Results are shown in Table 4, where #Sem. denotes the number of semantic tokens and #Pix. denotes the number of pixel tokens. All models are trained for one epoch on text-to-image datasets, and further continued on 1.4M high-quality image editing samples. For variants without pixel tokens, generation terminates after all semantic tokens are sampled, and images are decoded with TA-Tok's (Han et al., 2025) autoregres-

sive de-tokenizer. Evaluation is performed using GenEval (Ghosh et al., 2023) and DPG (Hu et al., 2024) for text-to-image generation, and ImgEdit (Ye et al., 2025b) for image editing.

The results reveal several key trends. First, using only pixel tokens yields the poorest scores, highlighting their limited alignment with language tokens. Second, incorporating semantic tokens consistently improves performance, demonstrating their role in bridging the gap between language and vision. Finally, when comparing variants that use the same number of semantic tokens, our approach of natively generating pixel tokens outperforms those that rely on external generative de-tokenizers on all text-to-image and editing benchmarks, with the advantage becoming more pronounced when the semantic sequence is short (e.g., 81 tokens). This configuration not only achieves the best overall performance but also increases the sequence length by only ∼7.9%, indicating that a small set of semantic tokens is necessary and sufficient for effective in-context cross-modal alignment.

**Effectiveness of the MoT Architecture.** Bridge adopts a Mixture-of-Transformers (MoT) architecture to develop visual generation ability on top of pre-trained understanding-oriented MLLMs. To assess the necessity of this design, we compare it with a straightforward dense architecture, where a single transformer backbone inherited from the same pre-trained MLLM is continually trained on a mixture of text-to-image, image understanding, and pure language data. As shown in Table 5, after continued training, the dense variant achieves a slightly higher DPG score but performs significantly worse on GenEval and multiple understanding benchmarks. This result suggests that generative training inevitably degrades the visual understanding ability of the base MLLM, whereas the MoT architecture effectively isolates and preserves understanding while adding generative capacity, validating the importance of our design.

**Token Routing Scheme.** In Bridge, visual generation ability is introduced by routing newly added generation image tokens to the generation expert, while all understanding image tokens and text tokens remain routed to the understanding expert. This design preserves the original comprehension capability of the base MLLM while extending it with generative capacity. To examine alternative routing strategies, we experiment with all combinations of routing for understanding image tokens and text tokens, and report results in Table 6. Note that generation image tokens are always routed to the generation expert, since they cannot be modeled by the frozen understanding expert. For cases where understanding image tokens are assigned to the generation expert, they are still encoded by the inherited continuous vision encoder to ensure consistency.

The results highlight several key trends. Our default routing scheme (row 1), where only generation tokens are handled by the generation expert, achieves the best overall balance: strong visual generation performance and excellent visual understanding ability. Moving understanding tokens into the generation expert (row 2) causes interference between semantic and pixel representations, leading to degradation in both understanding and generation benchmarks. Routing text tokens to the generation expert (row 3) is counterintuitive and unsurprisingly results in worse visual understanding. Finally, sending all tokens to the generation expert (row 4) collapses to the dense model setting, where limited capacity yields the weakest visual understanding performance. Interestingly, this variant also achieves the lowest GenEval score but slightly outperforms other schemes on DPG.

In summary, keeping understanding image tokens and text tokens routed to the understanding expert, while restricting the generation expert to newly introduced generation tokens, delivers the best overall trade-off between visual understanding and generation.

## 5 CONCLUSION

In this paper, we present Bridge, a pure autoregressive unified MLLM that augments pre-trained models with visual generative capability while rigorously preserving their visual understanding, enabling seamless multimodal understanding and generation within a single next-token prediction paradigm. Central to our approach is a semantic-to-pixel discrete visual representation that fuses compact high-level semantic tokens with fine-grained pixel tokens, achieving strong language alignment and high-fidelity synthesis with only a modest increase in sequence length. Extensive experiments across diverse benchmarks demonstrate competitive or superior performance in both understanding and generation, alongside reduced data requirements, shorter training time, and overall improved efficiency compared to prior unified MLLMs.

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

# Appendix

## A  OVERVIEW

- Section B:Additional Related Work
- Section C: Dataset
- Section D: Training Hyper-parameters
- Section E: Comparison Details
- Section F: More Visualization
- Section G: Limitation
- Section H: LLM Usage Disclosure

## B  ADDITIONAL RELATED WORK

**Visual Generation.**  Discrete autoregressive (AR) models generate images by sequentially predicting tokens, conditioned on class labels or text. Scalable AR approaches include next–scale prediction (Tian et al., 2024), randomized sampling (Yu et al., 2024b), continuous-token Fluid (Fan et al., 2024), tokenizer pre-alignment (Wang et al., 2025a), and large bitwise tokenization in Infinity (Han et al., 2024), with hybrid designs such as HART combining AR backbones with lightweight diffusion modules (Tang et al., 2024). Continuous diffusion models remain dominant for text-to-image generation, from cascaded (Ho et al., 2022) and score-based (Batzolis et al., 2021) diffusion to conditional text/image diffusion (Zhu et al., 2023; Graikos et al., 2024; Zheng et al., 2023) and recent inference-time scaling (Ma et al., 2025b). Seedream 3.0 further advances bilingual text-conditioned diffusion with timestep sampling and representation alignment (Gao et al., 2025). Recent comparisons show AR can match or surpass diffusion in scalability and quality (Sun et al., 2024). Unified generative frameworks aim to bridge class-conditioned, text-conditional, and visual conditioning within a single model, as demonstrated by OmniGen (Xiao et al., 2025), Janus (Wu et al., 2025b), TokenFlow (Qu et al., 2025b), and unified latent diffusion (Ma et al., 2023). These trends highlight a convergence of discrete AR and continuous diffusion paradigms toward versatile, multimodal image generation.

## C  DATASET

In Table 7, we summarize our datasets of different training stages and tasks. Now we will explain the detail of each dataset.

**Stage I: Unified Pretraining.**  In this stage, we collect and filter diverse image-text datasets and interleaved datasets to equip the model with fundamental image-text alignment capability.

- **LAION-5B-Filtered.** We apply aesthetic score ($>5$) and resolution ($>512$px) filtering to LAION-5B (Schuhmann et al., 2022), resulting in 385M samples. Then we use Qwen2.5-VL (Qwen et al., 2025) and InternVL3 (Zhu et al., 2025) to generate both short and long captions for each image.
- **Omnicorpus-Filtered.** Similar to LAION-5B-Filtered, we apply aesthetic score and resolution filtering to Omnicorpus (Li et al., 2024c) and collect 20M image-text interleaved samples. To enhance the image-text alignment, we use Qwen2.5-VL to recaption and refine each sample's text content, where each image is corresponding to a short paragraph.
- **Infinity-MM-Stage1-2 and Capsfusion.** We use the stage 1 and stage 2 data of Infinity-MM (Gu et al., 2024), which contains image-text pairs and general visual instruction tuning datasets. To balance the data ratio, we also incorporate 20M image-text pairs from Capsfusion (Yu et al., 2024c).
- **VINCIE** is a large-scale multi-turn image editing dataset annotated from videos.

Table 7: **Dataset Summary**. We list our datasets for Unified Pretraining, Continued Pretraining and Supervised Finetuning. T2I: Text-to-image. I2T: Image-to-text.

| Stage | Type | Data (Size) |
|---|---|---|
| Unified Pretrain | T2I | LAION-5B-Filtered (385M) |
| | I2T | Infinity-MM-Stage1-2 (37M), Capsfusion (20M) |
| | Interleave | Omnicorpus-Filtered (20M), VINCIE (9M) |
| Continued Pretrain | T2I | LAION-5B-Filtered (145M), Tar-Gen23M (23M), LAION-OCR-30M (30M), LAION-Face-4M (4M) |
| | I2T | Infinity-MM (20M), Capsfusion (10M) |
| | Interleave | Omnicorpus-Filtered (5M), VINCIE (9M) |
| Supervised Finetuning | T2I | JourneyDB-SD (1.4M), IN1K-SD (1.3M), DALLE3-SD (1M), Megalith-SD (3.5M), BLIP-3o (60K), ShareGPT-4o-Image (95K), Echo-4o-Image (180K), Text-QwenImage (300K) |
| | Edit | OmniGen2 (3.2M), OmniEdit (1.2M), AnyEdit (2.5M), GPTImgEdit (1.5M) |
| | I2T | Mammoth-VL (10M), Infinity-MM-Stage4 (1.8M) |
| | Interleave | LCT (200K) |

**Stage II: Continued Pretraining.** For this stage, we focus on enhancing the model's visual generation ability on high-quality datasets.

- **LAION-5B-Filtered.** we apply higher aesthetic score threshold ($>6$) and resolution ($>1024$px) filtering to LAION-5B, resulting in 145M samples.

- **OCR and Face.** We also leverage OCR and face detection models to curate text rendering and face datasets. Except the aesthetic score and resolution filters, an image will be preserved when it has at lease one text or human face. To suppress very challenging small face, we only keep images with human face $>32\times32$px.

- **Tar-Gen23M.** We also leverage Tar-Gen23M datasets (Han et al., 2025), which consists of 23M high-quality images generated by FLUX (Labs, 2024) using diverse text prompts (Deng et al., 2009; DrawThingsAI, 2024; Sun et al., 2023; Egan et al., 2024).

**Stage III: Supervised Finetuning.** Although the model has basic image generation capability after Stage I and Stage II training, the SFT stage is very crucial for improving the model's visual quality.

- **GPT-4o-Image Datasets.** Recently, several datasets generated by GPT-4o-Image are proposed, demonstrating strong instruction following and high image quality. These datasets include BLIP3o-60K (Chen et al., 2025a), ShareGPT-4o-Image (Chen et al., 2025b) and Echo-4o-Image (Ye et al., 2025a) for text-to-image generation, GPTImageEdit (Wang et al., 2025b) for image editing.

- **Seedream Datasets.** The amount of GPT-4o-Image datasets cannot meet our requirements for balanced training with other tasks (*e.g.*, I2T and Edit). Therefore, we curate million-scale synthetic datasets using Seedream 3.0 (Gao et al., 2025). We re-use prompts in previous text-to-image datasets such as JourneyDB (Sun et al., 2023) and DALLE3-Images (Egan et al., 2024) to create JourneyDB-SD and DALLE3-SD. We also leverage Qwen2.5-VL to generate detailed captions for existing image datasets such as ImageNet(Deng et al., 2009) and Megalith(DrawThingsAI, 2024), and then generate corresponding images using Seedream 3.0.

- **Text-QwenImage.** To enhance the model's text rendering quality, we curate Text-QwenImage-300K dataset. Given diverse prompts from existing datasets, we first transform them for the text rendering task using GPT-4o. For example, a prompt "*A boy and a dog is running*" will be augmented as "*A boy and a dog is running, with a banner in*

Table 8: **Training Hyper-parameters.** T2I: text-to-image. I2T: image-to-text. IL: interleave datasets. EDIT: image editing datasets.

| Stage | Unified Pretrain | Continued Pretrain | Supervised Finetuning |
|---|---|---|---|
| learning rate | 5e-5 | 1.25e-5 | 1.25e-5 |
| lr schedule | cosine | cosine | constant |
| optimizer | AdamW | AdamW | AdamW |
| optimizer params | $\beta_1$=0.9,$\beta_2$=0.999 | $\beta_1$=0.9,$\beta_2$=0.999 | $\beta_1$=0.9,$\beta_2$=0.999 |
| weight decay | 1e-4 | 1e-4 | 1e-4 |
| input resolution | 512 | 512 | 512 |
| warmup steps | 2000 | 2000 | 100 |
| total samples | 496M | 60M | 28M |
| global batch size | 1536 | 384 | 384 |
| gradient clip | 1.0 | 1.0 | 0.1 |
| data ratio | 14:2:1 (T2I:I2T:IL) | 7:2:1 (T2I:I2T:IL) | 4:3:2:1 (T2I:EDIT:I2T:IL) |

*the background says 'Best Friends Forever'*". Then we prompt QwenImage to generate the corresponding images.

- **Image Edit Datasets.** We collect image editing datasets from existing works, such as OmniGen2 (Wu et al., 2025d), OmniEdit (Wei et al., 2024), AnyEdit (Yu et al., 2024a) and GPTImgEdit (Wang et al., 2025b).

- **Interleaved Datasets.** We use the Long Context Tuning Dataset (LCT) (Guo et al., 2025) for high-quality image-text interleaved finetuning. LCT is primarily designed for long video generation. Here we only extra key frames of each shot to construct a multi-turn text-to-image dataset.

## D  TRAINING HYPER-PARAMETERS

We list training hyper-parameters in Table 8.

## E  COMPARISON DETAILS

We evaluate Bridge against a broad set of baselines, grouped by task type.

**Visual Understanding.** For visual understanding benchmarks, we compare against both understanding-only and unified models. Understanding-only baselines include LLaVA-v1.5 (Liu et al., 2024b), Qwen-VL (Bai et al., 2023b), LLaVA-NeXT (Liu et al., 2024c), DeepSeek-VL (Lu et al., 2024), and LLaVA-OneVision (LLaVA-OV) (Li et al., 2024a). Unified baselines include IL-LUME (Wang et al., 2024a), Chameleon (Team, 2024), LWM (Liu et al., 2024a), Emu3 (Wang et al., 2024b), Liquid (Wu et al., 2024a), UniTok (Ma et al., 2025a), VILA-U (Wu et al., 2024b), Janus-Pro (Chen et al., 2025c), TokenFlow-XL (Qu et al., 2025b), MetaMorph (Tong et al., 2024), Tar (Han et al., 2025), LMFusion (Shi et al., 2024), MetaQuery-XL (Pan et al., 2025), Show-o2 (Xie et al., 2025b), UniWorld-V1 (Lin et al., 2025), and BLIP3-o (Chen et al., 2025a).

**Visual Generation.** For visual generation, we compare against both generation-only and unified models. Generation-only baselines include SDXL (Podell et al., 2023), Playground v2.5 (Li et al., 2024b), Hunyuan DiT (Li et al., 2024d), DALLE3 (Lin et al., 2024), SD3-Medium (Esser et al., 2024), and SANA-1.5 (Xie et al., 2025a). Unified baselines include Chameleon (Team, 2024), LWM (Liu et al., 2024a), Emu3 (Wang et al., 2024b), SEED-X-13B (Ge et al., 2024), Transfusion (Zhou et al., 2024), ILLUME (Wang et al., 2024a), Janus-Pro-7B (Chen et al., 2025c), Tar (Han et al., 2025), MetaQuery-XL (Pan et al., 2025), Show-o2-7B (Xie et al., 2025b), BAGEL (Deng et al., 2025), UniWorld-V1 (Lin et al., 2025), and BLIP3-o-8B (Chen et al., 2025a).

**Instructional Image Editing.** We assess instructional editing performance on the ImgEdit bench-mark (Ye et al., 2025b), comparing against both specialized editing models—Instruct-P2P (Brooks et al., 2023), AnyEdit (Yu et al., 2024a), UltraEdit (Zhao et al., 2024), and Step1X-Edit (Liu et al., 2025)—and unified MLLMs, including BAGEL (Deng et al., 2025) and UniWorld-V1 (Lin et al., 2025).

## F  MORE VISUALIZATION

We provide more visualization results of our model in Figure 3.

## G  LIMITATIONS

While Bridge achieves strong performance in both visual understanding and generation, several limitations remain. First, since Bridge relies on discrete vision encoders from Han et al. (2025) and Sun et al. (2024), its ultimate performance is bounded by the representational capacity of these encoders. Second, the high compression inherent to discrete token representations makes it challenging to synthesize fine-grained details such as small text within images, a common limitation shared by most discrete token–based image generators. Finally, Bridge currently lacks the ability to generate images with arbitrary resolutions. Extending the framework to support variable-resolution synthesis is an important direction for future work.

## H  LLM USAGE DISCLOSURE

We used an external large language model as an assistive writing tool to help with phrasing, grammar, and clarity during manuscript preparation. The LLM did not contribute to the core research ideas, technical design, experiments, or the scientific content. All final writing was reviewed, edited, and approved by the authors, and the authors take full responsibility for any errors or content in the paper.

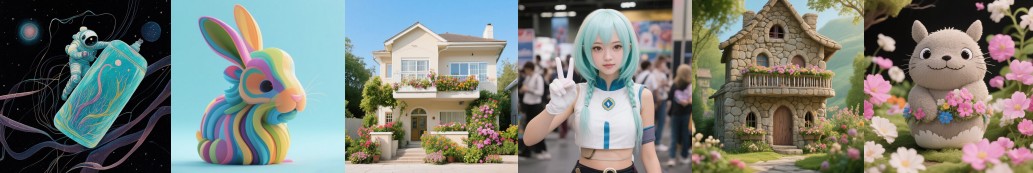

1. A charming 3D-rendered cartoon toucan with exaggerated features, styled in a whimsical yet detailed manner. The character features a disproportionately large, bright orange beak that dominates its face, complemented by a single oversized expressive brown eye. Its plumage combines classic black and white coloring - a fluffy white chest and belly area contrasts beautifully with sleek black feathers covering its back and wings. The bird sports a distinctive messy crest of navy blue feathers on top of its head, giving it a playful, disheveled appearance. The texturing is remarkably detailed, showing individual feathers and subtle variations in the plumage. The character stands on thin, sturdy legs with detailed scaled texture. The lighting setup creates depth and dimension, casting soft shadows that emphasize the bird's round, appealing form against a neutral gradient background. The overall design strikes a perfect balance between cartoon stylization and realistic texturing, making it suitable for animation or game character design.
2. ultra-detailed glass bottle terrarium featuring a miniature 3-story traditional Japanese villa, architectural photography style, precise architectural details with wooden beams and tiled roofs, delicate moss and tiny plants growing organically on the structure, ambient interior lighting casting warm glows through miniature shoji screens, the entire scene captured inside a clean cylindrical glass vessel, soft living room lighting enhancing glass reflections, tilt-shift photography effect emphasizing the miniature scale, photorealistic rendering with attention to glass refraction and natural materials
3. Gold and green mountain 3d illustration, in the style of fluid photography, orange and cyan, gold and cyan meticulous and detailed, Wang Ximeng, Northern Song Dynasty, Thousand-Mile Rivers and Mountains, Chinese landscape painting, traditional, vast and majestic, enchanting beauty, symbolism, glossy glass material, 4D render style, reflections.
4. bright colorful illustration of lake with view mountain and hype detailed sunset view
5. highly detailed full body jellyfish
6. generate anime looking Japanese cute girl futuristic looking

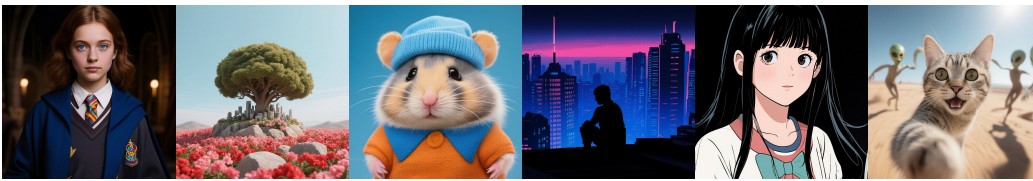

1. High-fashion wool felt handmade plush toy, cute chibi two-heads-tall style, pure white fluffy bunny, round face, upright ears with yellow star blush, sky-blue beady eyes with highlights, red polka-dot bib, standing on tiptoes, smiling. Felt-textured grass field, wildflowers, pure blue sky, giant yellow sun, warm soft lighting, HD.
2. "Paint an artistic oil painting featuring vibrant clay vases painted in bright colors, adorned with colorful botanical flowers, in a classical artistic style similar to the works of Da Vinci."
3. Fischer's lovebird, Galah Cockatoo, Lutino Ringneck Parakeet and other psittacidae species.
4. Create a fusion of Frieren from Frieren: Beyond Journey's End and Sailor Moon from Sailor Moon, with the character primarily resembling Frieren. She should have Frieren's calm and timeless expression, pointed elf ears, and signature pale blue hair styled in Sailor Moon's iconic twin buns with flowing strands. Her attire should merge Frieren's mage-like robes with subtle Sailor Moon-inspired celestial details, such as crescent moons and stars integrated into the design. She should wield a staff reminiscent of Frieren's wand, enhanced with a crescent moon motif at the top. Surround her with a soft, ethereal glow, symbolizing the fusion of Frieren's wisdom and magical mastery with Sailor Moon's celestial grace and heroism.
5. Holistic healing flowers medicinal pollinator nectar hive new beginning butterfly bees mushrooms
6. A child's drawing using crayons on a white piece of paper | a cityscape with tall, crooked buildings, stick figure people walking on the street, cars that look like rectangles with wheels, and a big smiling sun in the sky | Crayola, messy and lively.

1. Astronaut with fish tank is moving inside the galactic space suspended in the air with very fine and intertwined lines and acid watercolor and oil colors with strong contrast with different details and colors.
2. a female rabbit head, minimalistic colorful organic forms, energy, assembled, layered, depth, alive vibrant, 3D, abstract, on a light blue background
3. Blooming balcony goals. This house boasts a beautiful balcony overflowing with colorful flowers.
4. A beautiful cosplayer, portraying Ganyu, wore a cute expression and gave a victory sign. The background was a comic convention, captured with a Canon camera, using portrait focus.
5. Fairytale vibes. This stone house with its flower-filled balcony looks like it's straight out of a storybook.
6. cute picture with totoro and flowers

1. a girl with blue eyes, brown straught hair dressed in gryfindor hogwarts robes
2. A fully detailed city on a large tree in the middle of a plain full of red and pink flowers and pieces of rock
3. A hamster pawn wearing a blue and orange shirt. The hamster is wearing a hat. The hamsters must resemble those from Albert Heijn.
4. 8 bit style man silhouette sitting on skyscraper night scene, nostalgic synthwave colro scheme, but a little lonely
5. A girl with long straight hair in 80's anime vintage style
6. A Silver Tabby British Shorthair cat takes a selfie close-up with wide, surprised eyes, ears pinned back in panic, and a mischievous, smiling grin. In the background, three Paul-style aliens are in fast pursuit—clearly visible through motion blur and cinematic bokeh. The scene features dynamic, fast-action movement and is shot with a landscape fisheye lens in a desert setting.

Figure 3: **More Visualization** of our model on text-to-image generation.

