# OpenReview forum: "Growing Visual Generative Capacity for Pre-Trained MLLMs"
_ICLR.cc/2026/Conference — ICLR 2026 Conference Withdrawn Submission_

### Official Review · Reviewer_cNwT · 2025-10-31

**Soundness:** 2
**Presentation:** 3
**Contribution:** 2
**Rating:** 4
**Confidence:** 5

**Summary:**

This paper proposes Bridge, a purely autoregressive unified MLLM based on a mixture-of-transformer architecture. Bridge performs both visual understanding and generation within a single next-token prediction framework. Furthermore, it introduces a semantic-to-pixel discrete representation, which places semantic tokens before pixel tokens to enable semantic-aware generation.

**Strengths:**

1. The paper is well-written and easy to follow.
2. The experiments and ablation studies are detailed and comprehensive.

**Weaknesses:**

1. The authors claim to have proposed a semantic-to-pixel representation to achieve semantic alignment. However, the approach relies on TA-Tok to provide semantic tokens, which was originally introduced by Tar and is not a novel contribution of this paper. The main design change appears to be replacing the detokenizer in Tar with a single transformer that directly generates pixel tokens — a relatively minor modification. Moreover, the authors do not clearly demonstrate the advantages of this change. The unified transformer is also larger than the lightweight detokenizer, which may increase generation latency.
2. I do not consider the use of a mixture-of-transformer architecture to extend a pretrained understanding MLLM into a unified model to be a novel design. The authors themselves list many related works adopting similar ideas. Given that there are already separate modules for understanding and generation that share attention, the paper does not convincingly explain why adopting a purely autoregressive next-token prediction framework offers advantages over existing flow-matching designs.
3. Regarding the image editing experiments, the authors mention using OmniGen2 data but do not report its results on the ImgEdit benchmark.

**Questions:**

1. I noticed that visual understanding data is also used during training. Since a frozen pretrained MLLM is leveraged for visual understanding, it is unclear why this part of the data is needed.

---

### Official Review · Reviewer_gzZ5 · 2025-11-02

**Soundness:** 2
**Presentation:** 2
**Contribution:** 2
**Rating:** 4
**Confidence:** 4

**Summary:**

This paper introduces Bridge, a pure autoregressive unified multimodal large language model (MLLM) designed to perform both visual understanding and generation. The core idea is to augment a pre-trained MLLM with generative capabilities without compromising its original strengths. For image generation, this paper proposes a "semantic-to-pixel" discrete visual representation, where a short sequence of high-level semantic tokens is followed by a longer sequence of fine-grained pixel tokens, all predicted within next-token prediction framework.

**Strengths:**

1. The "semantic-to-pixel" representation uses appropriate semantic tokens and token arrangements to achieve both effectiveness and efficiency.
2. The comparison against a "dense" architecture (Tab.5) clearly validates the MoT design, while the token routing experiments (Tab.6) further justify the chosen architecture. These detailed analyses significantly enhance the credibility of the paper's conclusions.

**Weaknesses:**

1. The core MoT architecture, featuring separate QKV and FFN blocks for different experts that interact via a shared attention, is highly reminiscent of the dual-branch architecture used in BAGEL. The difference with BAGEL is a purely AR loss v.s. flow matching loss. Furthermore, it is empirically debaTab.whether a pure AR loss is superior to diffusion or flow-based losses for high-fidelity visual generation; the latter often demonstrate stronger performance in image quality and detail.
2. The DPG experimental results in Tab.2 are a significant concern. This benchmark is crucial for evaluating a model's ability to follow long and complex textual instructions, a key aspect of advanced text-to-image generation. The Tab.is notably missing performance results for several key unified models that employ denoising or flow-based training (e.g., BAGEL, MetaQuery, UniWorld, BLIP3-o).
3. It is desirable a direct, side-by-side qualitative comparisons with other unified models, especially those using diffusion or flow-matching objectives. Showing only the model's own outputs (Fig.1 and Appendix) is insufficient for a rigorous evaluation. The visual quality of AR models can differ significantly from diffusion models (e.g., in texture or global coherence), and a direct comparison would be highly informative.

**Questions:**

Please refer to weakness section.

---

### Official Review · Reviewer_FSuZ · 2025-11-02

**Soundness:** 2
**Presentation:** 2
**Contribution:** 2
**Rating:** 2
**Confidence:** 4

**Summary:**

This paper proposes Bridge, a unified multimodal large language model that trained purely with next-token prediction. For discrete image generation, the paper introduces a “semantic-to-pixel” representation: prepend a short sequence of semantic tokens to pixel-level tokens, claiming better language alignment and detail fidelity with only ~7.9% token overhead.

**Strengths:**

1.	The hard routing and frozen understanding branch offers a pragmatic way to avoid catastrophic degradation. The comparison against a "dense" architecture in Tab.5 clearly validates the MoT design, while the token routing experiments in Tab.6 further justify the chosen architecture.
2.	The comparison among “only pixel,” “only semantic,” and “semantic+pixel” is informative; the finding that a small number of semantic tokens helps alignment with a minor sequence-length cost is practically useful.

**Weaknesses:**

1.	Architectural novelty relative to BAGEL is limited; core design overlaps substantially, such as the MoT-style dual branches with isolated QKV/FFN per expert and shared attention, overlapping the core design substantially
2.	The idea of placing a short semantic token prefix before pixel tokens is natural, especially under causal attention. The role of CoT claimed in this article needs more evidence to prove, such as the attention distribution among tokens.
3.	The interaction between experts (“both experts share unified causal attention across all tokens” vs “hard routing” of tokens) is not fully clear. It needs explicit layer-by-layer description, parameter tying, memory/compute overhead, and how generation tokens condition the understanding branch during image-to-text tasks (mechanism of cross-branch conditioning).
4.	As a core benchmark for long-instruction adherence, DPG requires broad and balanced baseline coverage. The absence of denoising/flow-based unified models such as BAGEL, BLIP-3o and UniWorld, in Tab.2 weakens the paper’s claim that a pure autoregressive approach remains competitive in complex instruction scenarios. Please include these methods under the same evaluation protocol and explicitly report the use of any prompt rewriting to ensure fairness and credibility.

**Questions:**

see weaknesses

---

### Official Review · Reviewer_1o73 · 2025-11-02

**Soundness:** 2
**Presentation:** 3
**Contribution:** 3
**Rating:** 6
**Confidence:** 4

**Summary:**

This paper proposes Bridge, a pure autoregressive unified multimodal LLM that adds generative capacity onto a pre-trained visual understanding MLLM by using a Mixture-of-Transformers (MoT) dual-expert architecture. It introduces a semantic-to-pixel discrete visual representation (short sequence of semantic tokens followed by pixel tokens) to improve alignment with language while retaining pixel fidelity.

**Strengths:**

1. comprehensive comparison with existing works.
2. Practical focus on efficiency.
Limiting token length increase is a sensible way to avoid massive extra compute/data requirements. This paper emphasizes the use of less data and shorter training time compared to some unified baselines.
3. Comprehensive ablation studies.

**Weaknesses:**

1. The approach relies on TA-Tok for semantics and LlamaGen-VQGAN for pixels; the model’s ceiling is therefore tied to those encoders’ capacity. The paper acknowledges this limitation, but more analysis of encoder failures (and ablation with alternatives) would strengthen claims.
2. Discrete visual representations naturally have lower image quality than continuous ones (hence Bridge underperforms BAGEL, which uses a similar MoT architecture but diffusion).

**Questions:**

1. See "weaknesses" 1
2. Can you add some compute/memory profiling comparing MoT vs dense to justify cost/benefit?

---

### Note · Authors · 2025-11-14

I have read and agree with the venue's withdrawal policy on behalf of myself and my co-authors.